# Statistical Approaches to Identify Pairwise and High-Order Brain Functional Connectivity Signatures on a Single-Subject Basis

**DOI:** 10.3390/life13102075

**Published:** 2023-10-18

**Authors:** Laura Sparacino, Luca Faes, Gorana Mijatović, Giuseppe Parla, Vincenzina Lo Re, Roberto Miraglia, Jean de Ville de Goyet, Gianvincenzo Sparacia

**Affiliations:** 1Department of Engineering, University of Palermo, 90128 Palermo, Italy; laura.sparacino@unipa.it (L.S.); luca.faes@unipa.it (L.F.); 2Faculty of Technical Sciences, University of Novi Sad, 21102 Novi Sad, Serbia; gorana86@uns.ac.rs; 3Radiology Service, IRCCS-ISMETT, 90127 Palermo, Italy; gparla@ismett.edu (G.P.); rmiraglia@ismett.edu (R.M.); 4Neurology Service, IRCCS-ISMETT, 90127 Palermo, Italy; vlore@ismett.edu; 5Department for the Treatment and Study of Pediatric Abdominal Diseases and Abdominal Transplantation, IRCCS-ISMETT, 90127 Palermo, Italy; Jdeville@ismett.edu; 6Radiology Service, BiND, University of Palermo, 90128 Palermo, Italy

**Keywords:** single-subject analysis, functional connectivity, high-order interactions, surrogate data analysis, bootstrap validation

## Abstract

Keeping up with the shift towards personalized neuroscience essentially requires the derivation of meaningful insights from individual brain signal recordings by analyzing the descriptive indexes of physio-pathological states through statistical methods that prioritize subject-specific differences under varying experimental conditions. Within this framework, the current study presents a methodology for assessing the value of the single-subject fingerprints of brain functional connectivity, assessed both by standard pairwise and novel high-order measures. Functional connectivity networks, which investigate the inter-relationships between pairs of brain regions, have long been a valuable tool for modeling the brain as a complex system. However, their usefulness is limited by their inability to detect high-order dependencies beyond pairwise correlations. In this study, by leveraging multivariate information theory, we confirm recent evidence suggesting that the brain contains a plethora of high-order, synergistic subsystems that would go unnoticed using a pairwise graph structure. The significance and variations across different conditions of functional pairwise and high-order interactions (HOIs) between groups of brain signals are statistically verified on an individual level through the utilization of surrogate and bootstrap data analyses. The approach is illustrated on the single-subject recordings of resting-state functional magnetic resonance imaging (rest-fMRI) signals acquired using a pediatric patient with hepatic encephalopathy associated with a portosystemic shunt and undergoing liver vascular shunt correction. Our results show that (i) the proposed single-subject analysis may have remarkable clinical relevance for subject-specific investigations and treatment planning, and (ii) the possibility of investigating brain connectivity and its post-treatment functional developments at a high-order level may be essential to fully capture the complexity and modalities of the recovery.

## 1. Introduction

Network representation is an effective tool for comprehending complex systems and the interactions among their distinct components. In neuroscience, network analysis is valuable for identifying patterns of connectivity and communication within the brain [1,2]. Over the last few decades, brain connectivity has been extensively investigated [2,3,4,5,6,7,8], with the aim to disentangle and understand the underlying mechanisms of resting-state scenarios, as well as of different cognitive and perceptual tasks necessitating a co-ordinated flow of information, which, in turn, changes according to the strength and pattern of oscillatory synchrony within and between networks of functionally specialized brain regions. Modern neuroscience employs a variety of electrophysiological and neuroimaging techniques to explore brain connectivity associated with both normal and neuro-pathologic functions, such as functional magnetic resonance imaging (fMRI), widely used to quantify hemodynamic changes (i.e., spontaneous blood oxygen level-dependent—BOLD—signal fluctuations) following the activation of specific brain areas [2,9,10,11]. Specifically, resting-state fMRI (rest-fMRI) is a novel neuroimaging technique that explores the intrinsic brain functional architecture, or connectome, associated with both normal and neuropathologic functions by examining resting-state networks (RSNs) in the resting or relaxed state [10,12,13,14,15]. Despite ongoing standardization efforts for rest-fMRI, there is evidence suggesting its potential to provide valuable insights into the organization of brain neuronal networks in routine clinical settings [10].

Several measures have been developed to examine functional connectivity in the brain, probing the intricate interactions between the elements of cerebral networks. Approaches exploring pairwise connectivity patterns, such as mutual information (MI) [16], are easily applicable, require little computational effort, and offer a straightforward interpretation of the findings. Although highly effective [17,18,19,20], these methods are inherently restricted by the constructional requirement that every interaction must be between two elements, i.e., pairwise. However, there is mounting evidence that such measures cannot fully capture the interplay among the multiple units of a complex system [18,21,22]. Consequently, recognizing and modeling high-order functional structures, which are characterized by statistical interactions involving more than two network units, has become a crucial and evolving area of complex systems research [1,21,22]. In network neuroscience, high-order interactions (HOIs) have been suggested to be the fundamental components of complexity and functional integration in brain networks [23], and they are proposed to be linked to emergent mental phenomena and consciousness [24]. Nevertheless, in spite of their promising significance, the investigation of HOIs in the brain is a relatively unexplored domain. Given that these interactions are not typically accessible through the well-established pairwise measures of functional connectivity network analysis, their study has often been limited by the lack of formal tools, as well as by the involvement of inherent computational and combinatorial challenges. While many different information theoretic metrics have been proposed throughout the years, all attempting to capture the information shared by triplets of random variables or processes [25,26,27,28], a recent work [29] suggests the potential use of information theory for identifying HOIs in multivariate systems, as well as for distinguishing between qualitatively distinct modes of information sharing, i.e., redundancy and synergy. These two general concepts refer to the nature of the interactions among the multiple units of a complex system (e.g., the brain, the human body, the global climate, or any financial system) [21,22,30]. Specifically, redundancy refers to group interactions that can be explained by the communication of subgroups of variables, thus pertaining to information that is replicated across numerous elements of the complex system, i.e., common information or patterns being shared: observing subsets of elements can resolve uncertainty across all the other elements of that system. Conversely, synergistic information sharing takes place when the joint state of three or more variables is necessary to resolve uncertainty arising from statistical interactions that can be found collectively in a network but not in parts of it considered separately. Synergy is a potentially intriguing phenomenon as it reflects the ability of the human brain to generate new information by combining the interplay of anatomically distinct but functionally connected brain areas. A measurement of this quantity in the context of HOIs comes from O-information (OI) [29], which provides an overall evaluation of whether a system is dominated by redundancy or synergy.

The statistical validation of brain connectivity metrics is essential to detect the significance of the associations between pairs or groups of network nodes, as well as to investigate their variability across subjects or experimental conditions. Indeed, it is commonly known that spurious connectivity patterns may arise even in the case of complete uncoupling between the analyzed signals due to structural misunderstandings [31], finite data size effects, or acquisition/computation errors that show an estimated value that deviates from the true connectivity value [32]. Furthermore, while such validation is typically performed at the level of subject groups, in clinical practice, where the goal is to optimize the individual treatment plan and look into the effects of interventions on a single patient [2,10,33], statistical analyses should be focused on subject-specific differences between experimental conditions. Indeed, the increasing demand for personalized neuroscience necessitates drawing conclusions from connectivity metrics obtained from individual recordings of brain signals. In order to address this, methods to determine the accuracy (confidence limits) of individual estimates of the considered indexes are still necessary. This is especially important as the accuracy of estimates may vary over time and depend on factors such as the individual (patho) physiologic state. The absence of confidence limits or error bounds on estimates may lead to biased clinical decisions, making it imperative to ensure a reliable assessment of the patient’s underlying condition.

By taking the previous concepts into account, the rationale of this work is to emphasize the importance of single-subject analysis to investigate brain connectivity in different physiopathological states, as well as the need to exploit novel high-order measures capturing the properties of complex brain networks as a whole. To this purpose, we propose a single-subject-based approach to statistically assess pairwise and high-order connectivity patterns in brain networks, investigated respectively through the MI and OI measures estimated in the framework of linear parametric regression models based on the utilization of surrogate and bootstrap data analyses. Specifically, surrogate time series, mimicking the individual properties of the original series but being otherwise uncoupled [34], are exploited to assess whether the dynamics of two putatively interacting nodes are significantly coupled or not, while the bootstrap technique [35] is employed to generate confidence intervals that allow the significance assessment of HOIs, as well as the comparison of individual estimates of the considered indexes across different experimental conditions. The approach is validated on single-subject recordings of multivariate fMRI signals, confirming that the single-subject analysis of network connectivity can provide detailed information on the brain functions across different physiopathological states. In clinical practice, the utilization of the proposed single-subject statistical validation approach through surrogate and bootstrap data analyses is essential to focus on subject-specific interventions and treatments, as well as to ensure a reliable assessment of the individual’s underlying condition. Moreover, our results support the application of multivariate information measures on a single-subject basis to unveil synergistic “shadow structures” emerging from resting-state brain activity, missed by bivariate functional connectivity approaches, which indeed reveal redundancy-dominated correlations and do not provide an overall map of the statistical structure of the network, as clearly demonstrated in [36,37,38]. This novel combined exploitation of complex network analysis through high-order measures and single-subject statistical validation approaches is the core of our work, and this has the great potential to detect subject-specific changes of complex brain connectivity patterns in different physiopathological states. In order to visualize these concepts and better understand the rationale behind our study, we refer the reader to Figure 1.

## 2. Methods and Materials

This section reports the mathematical details of the methodology proposed for the assessment and statistical validation of subject-specific pairwise and high-order brain functional connectivity measures (Section 2.1). A schematic workflow of the proposed methodology is reported in Figure 1, illustrating the rationale of the approach and its main steps. The approach is then validated in a simulation example and applied to a clinical dataset (Section 2.2).

### 2.1. Methods

Let us consider a network of *Q* random variables, S = $\left\{S_{1},\ldots ,S_{Q}\right\}$. The multiple interactions between *N* variables taken from the set $\left\{S_{1},\ldots ,S_{Q}\right\}$ can be investigated by means of a static analysis of multiple realizations of these variables, available in the form of multiple time series; in this analysis, temporal correlations are disregarded, and only zero-lag effects are taken into account. This is typically carried out in the field of brain functional connectivity [9,14], which, indeed, quantifies the temporal dependency of neuronal activation patterns of anatomically separated brain regions [19]. In this context, well-established measures defined in the framework of information theory can be exploited to study the interactions between pairs (i.e., N=2) and/or groups of variables (i.e., N=3,…,Q) taken from S.

#### 2.1.1. Connectivity Measures: Pairwise and High-Order Approaches

The pairwise link between two variables, $S_{i}$ and $S_{j}$, taken from the set $\left\{S_{1},\ldots ,S_{Q}\right\}$, with i,j=1,…,Q;i≠j can be investigated by means of the measure of mutual information (MI), which quantifies the information shared between the two variables based on the concept of Shannon entropy [16,39]. In the framework of information theory, the MI is defined as
(1)$$I\left(S_{i};S_{j}\right)=H\left(S_{i}\right)-H\left(S_{i}|S_{j}\right),$$
where H(·) denotes the entropy of a single variable, measuring the amount of information carried by that variable, while H(·|·) denotes conditional entropy, measuring the information carried by the first variable when the second is known [40].

Nevertheless, in complex networks of interacting variables, HOIs involving more than two nodes at a time often emerge and display patterns that cannot be detected using pairwise measures. HOIs can be assessed by means of O-information (OI), a novel theoretic measure of information that generalizes the concept of MI to groups of variables [29]. The OI of *N* random variables taken from the set $\left\{S_{1},\ldots ,S_{Q}\right\}$ is defined as
(2)$$\Omega \left(S^{N}\right)=\Omega \left(S_{-j}^{N}\right)+\Delta \left(S_{-j}^{N};S_{j}\right),$$
where $S^{N}=\left\{S_{i_{1}},\ldots ,S_{i_{N}}\right\}$ ($i_{1},\ldots ,i_{N}\in \left\{1,\ldots ,Q\right\},N\leq Q$) is the analyzed group of random variables, $S_{-j}^{N}=S^{N}\setminus S_{j}$ is the subset of random variables where $S_{j}$ is removed (j∈$\left.\left\{i_{1},\ldots ,i_{N}\right\}\right)$, and where the quantity
(3)$$\Delta \left(S_{-j}^{N};S_{j}\right)=\left(2-N\right)I\left(S_{-j}^{N};S_{j}\right)+\sum_{\begin{array}{c} m=1\\ m\neq j \end{array}}^{N}I\left(S_{-mj}^{N};S_{j}\right)$$
is the variation of the OI obtained with the addition of $S_{j}$ to $S_{-j}^{N}$, being $S_{-mj}^{N}=S^{N}\setminus \left\{S_{j},S_{m}\right\}$ (in the following, referred to as Δ(OI)). Importantly, the sign of Δ(OI) can take on both positive values, meaning that the information brought by $S_{j}$ to $S_{-j}^{N}$ is prevalently redundant, and negative values, meaning that the influence of $S_{-j}^{N}$ on $S_{j}$ is prevalently synergistic. Then, the system is redundancy-dominated when *OI*
>0, and synergy-dominated when *OI*
<0; if *OI*
=0, synergy and redundancy are balanced in the analyzed network. Moreover, since $\Omega (S^{2})=0$ for any pair of variables, when N=3 random variables are considered, i.e., $S=\left\{S_{i},S_{k},S_{j}\right\}$, the OI in (Equation 2) reduces to the OI increment:(4)$$\Delta \left(S_{i},S_{k};S_{j}\right)=-I\left(S_{j};S_{i},S_{k}\right)+I\left(S_{j};S_{i}\right)+I\left(S_{j};S_{k}\right),$$
which, in turn, coincides with the well-known interaction information [41,42], measuring the balance between synergy and redundancy when a target variable is added to a bivariate source vector process.

#### 2.1.2. Computation through Linear Parametric Regression Models

The computation of the pairwise and high-order connectivity measures defined above requires an approach to compute the MI between random variables. Assuming that the observed variables are stationary and have a joint Gaussian distribution, the analysis can be performed by exploiting linear parametric regression models. Furthermore, it was demonstrated that regression methods show greater sensitivity to the coupling between the observed variables, with better performances with respect to other, more sophisticated approaches, requiring specific assumptions about the underlying model of the relationship [8]. Specifically, two generic zero-mean vector variables, X and Y, containing respectively $t_{1}$ and $t_{2}$ scalar variables, are related by the following linear regression model:(5)X=AY+U,
where X is predicted using a $t_{1}\times t_{2}$ coefficient matrix, A, which weights the regressors of Y, and $U={\left[U_{1}\ldots U_{t_{1}}\right]}^{⊺}$ is a vector of $t_{1}$ zero-mean white noises.

In the multivariate setting introduced in Section 2.1.1, the vectors X and Y can be identified properly by selecting variables from the set $\left\{S_{1},\ldots ,S_{Q}\right\}$ to investigate the pairwise (i.e., N=2) or high-order (i.e., N=3,…,Q) interactions between those variables of the network. Specifically, the MI can be assessed from the linear regression model in (Equation 5), exploiting the relation between entropy and variance that is valid for Gaussian variables. In this case, the entropy of the predicted variable can be expressed as $H\left(X\right)=$$\frac{1}{2}\log\left({\left(2\pi e\right)}^{t_{1}}\left|\Sigma_{X}\right|\right)$, while the conditional entropy of the predicted variable given the predictor is given by $H\left(X|Y\right)=\frac{1}{2}\log\left({\left(2\pi e\right)}^{t_{1}}\left|\Sigma_{U}\right|\right)$, where $\Sigma_{X}$ is the $t_{1}\times t_{1}$ covariance matrix of the predicted vector variable, $\Sigma_{U}$ is the $t_{1}\times t_{1}$ covariance matrix of the prediction errors, and |·| is the matrix determinant. This allows us to rewrite the MI in (Equation 1) as
(6)$$I\left(X;Y\right)=\frac{1}{2}\log\left(\frac{|\Sigma_{X}|}{|\Sigma_{U}|}\right).$$

This measure is symmetric and monotonically related to the cross-correlation between X and Y [43]. As such, it can be exploited to quantify the pairwise connectivity between two random variables, $S_{i}$ and $S_{j}$, as well as the high-order interactions between $S_{j}$ and $S_{-mj}^{N}$ (or, analogously, $S_{-j}^{N}$), taking the role of X and Y in (Equation 5), respectively.

#### 2.1.3. Statistical Validation

This section presents the use of surrogate and bootstrap data analyses to statistically validate the proposed measures of pairwise and high-order interactions. Validation is performed at the level of individual realizations of the observed variables $\left\{S_{1},\ldots ,S_{Q}\right\}$, obtained in the form of the set of simultaneously measured time series $s_{i}=\left\{s_{i}\left(1\right),\ldots ,s_{i}\left(M\right)\right\}$, where i=1,…,Q and *M* represent the length of the time series.

##### Surrogate Data Analysis

The method of surrogate data [34] is employed to obtain a threshold for zero pairwise connectivity, setting a significance level for the MI measure.

Specifically, a linear model as in (Equation 5) is first identified on the time series {x,y}, with $x=s_{j}$ and $y=s_{i}$, i,j=1,…,Q;i≠j. Then, estimates of the MI, denoted as I(x;y), are obtained using (Equation 6), with $t_{1},t_{2}=1$. Afterward, a surrogate time series that preserves the individual linear correlation properties of two series but destroys any correlation between them is obtained through the iterative amplitude-adjusted Fourier transform (iAAFT) procedure [44], which represents an advancement over the Fourier transform (FT) method. It generates surrogate time series by computing the FT of the original series, substituting the Fourier phases with random numbers uniformly distributed between 0 and 2π, and finally performing the inverse FT. Then, to address the main limitation of the FT method, which consists of a distortion of the amplitude distribution when such a distribution is not Gaussian, an iterative procedure is implemented, which alternately ensures that the surrogate series maintains both the same power spectrum and amplitude distribution as the original series. This procedure is repeated $N_{s}$ times to obtain the set of surrogate series $x^{s}$ and $y^{s}$, $s=1,\ldots ,N_{s}$. The MI is then estimated on each surrogate pair, yielding the distribution $I^{s}\left(x;y\right)$, from which the significance threshold $I^{\alpha }\left(x;y\right)$ is derived taking the $100{\left(1-\alpha \right)}^{th}$ percentile. Finally, the original MI value is deemed as statistically significant if $I\left(x;y\right)>I^{\alpha }\left(x;y\right)$. In this work, $N_{s}=300$ surrogate pairs were generated to assess the existence of significant pairwise connectivity.

##### Bootstrap Data Analysis

The bootstrap method [35] is used to identify confidence intervals for the MI and OI measures. For this purpose, the block bootstrap data generation procedure [45] is followed to generate, starting from the time series $s_{i}$ (i=1,…,Q), $N_{b}$ bootstrap pseudo-series $s_{i}^{b}=\left\{s_{i}^{b}\left(1\right),\ldots ,s_{i}^{b}\left(M\right)\right\}$, $b=1,\ldots ,N_{b}$, which maintain all the features of the original time series, i.e., individual and coupling properties. The procedure creates the bootstrap pseudo-series $s_{i}^{b}$ by joining together $k=\frac{M}{L}$ non-overlapping blocks chosen randomly from the set $\left\{B_{1},\ldots ,B_{k}\right\}$, where *L* is the size of each block, $B_{m}=\left\{s_{i}\left(m\right),\ldots ,s_{i}\left(m+L-1\right)\right\}$, and *m* is chosen randomly from the set {1,…,M−L+1}.

Then, the MI is recomputed from the new, full-size bootstrap series $x^{b}=s_{i}^{b}$ and $y^{b}=s_{j}^{b}$ to get the estimate $I^{b}\left(x;y\right)$, while the OI is recomputed at each order *N* from the new, full-size bootstrap series $s_{i_{1}}^{b},\ldots ,s_{i_{N}}^{b}$ ($i_{1},\ldots ,i_{N}\in \left\{1,\ldots ,Q\right\},N\leq Q$) to get the estimate $\Omega^{b}\left(s^{N}\right)$. The procedure is iterated for $b=1,\ldots ,N_{b}$ to construct bootstrap distributions. In this work, $N_{b}=300$ bootstrap repetitions were generated to identify confidence intervals for the investigated measures.

##### Statistical Significance of HOIs

The confidence intervals of the bootstrap distributions obtained, as described above, can be exploited to check the statistical significance of the absolute OI values in (Equation 2) and the OI increments in (Equation 3). Specifically, when a given bootstrap distribution comprises a zero threshold at the α significance level, i.e., if the zero value is below the $100{\left(1-\frac{\alpha }{2}\right)}^{th}$ and above the $100{\left(\frac{\alpha }{2}\right)}^{th}$ percentile of that distribution, the corresponding OI measure is deemed as not statistically significant. Moreover, the proposed bootstrap method allows to check whether the OI increment in (Equation 3), due to the addition of a putative target $s_{j}$ to a given lower-order group of variables (referred to as multiplet) $s_{-j}^{N}$ of order N−1, with N=4,…,Q, is significant or not, i.e., if the OI value computed for $s^{N}$ significantly differs from the same measure computed for $s_{-j}^{N}$. In order to do that for each order, the bootstrap distributions of the OI computed for all the multiplets of two consecutive orders *N* and N−1 can be exploited. When fixing the multiplet at order *N*, all the roots of that multiplet at the preceding order N−1, i.e., lower-order multiplets, for which the elements are all contained in the high-order multiplet, are identified. Then, for each root, the lower-order and the high-order bootstrap distributions are compared by means of the parametric Student *t*-test for unpaired data. Finally, the corresponding OI increment is deemed significant when the difference between the two distributions is significant at the α significance level according to the statistical test.

##### Statistical Significance of the Difference between Experimental Conditions

When the MI and OI measures are computed on a single-subject basis in two different experimental conditions, the bootstrap distributions can be employed to assess the significance of the difference between the two conditions through a statistical test. To this end, the bootstrap data generation procedure is executed in both the analyzed experimental conditions, and the parametric Student *t*-test for unpaired data is then employed to assess the statistical significance of the difference between pairs of bootstrap distributions for a given measure. Note that, in this work, a significance level α=0.05 was used both to compute the confidence intervals of the surrogate and bootstrap distributions as well as to perform statistical tests.

### 2.2. Materials

This section introduces the application of the proposed methodology to a simulated network of stochastic Gaussian variables (Section 2.2.1) and to single-subject recordings of the rest-fMRI signals acquired in a clinical case study of a pediatric patient suffering from hepatic encephalopathy (Section 2.2.2).

#### 2.2.1. Simulation Example

The framework for the computation of pairwise and high-order interactions is illustrated, making use of a theoretical example of simulated linear regression models, for which the MI and OI measures are computed directly from the known model parameters. This simulation is exploited to show that high-order measures can be used to highlight the emergence of the patterns of interaction among groups of variables that cannot be traced from pairwise connections alone, as well as evidencing the presence of circuits dominated by synergy or redundancy. Moreover, we show how the methods of surrogate and bootstrap data analysis can help to disregard nonsignificant interaction pathways among the variables, thus allowing us to focus only on specific connectivity links within the network.

The simulation is focused on the analysis of Q=7 random variables, with the network structure and interdependencies specified in Figure 2A. The parameters a(i), i=1,…,6, quantifying the pairwise relationships between the observed variables, are chosen in the range [0.95–1], setting a(1)=a(2)=0.99, a(3)=a(6)=1 and a(4)=a(5)=0.95. The parameter range was suitably selected to impose a clear coupling between the observed variables since lower values of the parameters (i.e., a(i)<0.95, i=1,…,6) would lead to lower and, hence, possibly nonsignificant values of the MI measure computed between those variables. The network is designed to simulate three zero-mean random noises, $X_{1}$, $X_{4}$, and $X_{5}$, with unit variance, for which the sink (commonly called common child) is the node $X_{2}$. Then, through a chain structure, $X_{2}$ converges into the node $X_{3}$, which, in turn, acts as a common driver for the nodes $X_{6}$ and $X_{7}$ (Figure 2A). From the resulting network $S=\{X_{1},\ldots ,X_{7}\}$, implemented via time series realizations of M=500 points, the time-domain MI between the pairs of variables was estimated, as in (Equation 6); then, its significance was assessed by applying the method of surrogate data and evaluating the existence of each pairwise link, as described in Section 2.1.3 (Figure 2B). The OI was computed, as in (Equation 2), for all the possible multiplets of orders N=3,…,7, and deemed as significant when the OI distributions, computed via bootstrap data analysis, did not comprise the zero level (Figure 2C). Moreover, the values of the OI increment were computed as in (Equation 3), at each order *N* and for each target $X_{j}$ within the selected multiplet $X^{N}$ of that order, and these were deemed as significant when the information brought by $X_{j}$ to $X^{N}$ was statistically significant according to the test discussed in Section 2.1.3 (Figure 2D).

#### 2.2.2. Application to Brain Networks

The proposed framework is applied to a clinical case involving one pediatric patient treated at our hospital, the IRCCS—ISMETT (Scientific Institute for Research, Hospitalization and Healthcare—Mediterranean Institute for Transplantation and Advanced Specialized Therapies), Palermo, Italy, with a cavernous transformation of the portal vein, an obstruction also known as portal cavernoma, which is a common cause of portal hypertension in children. Even in cases where liver function appears normal, this disease can result in the development of hepatic encephalopathy (HE) [46] due to the presence of portal-systemic shunts, which cause an increase in plasma ammonia levels and toxic brain catabolites deposition in the globi pallidi. HE is a serious condition that can have a profound impact on the patient’s ability to perform daily tasks, causing psychomotor sluggishness, attention deficits, and a decline in fine motor performance. Although HE is currently diagnosed using psychometric and electrophysiological examinations, the administration and interpretation of psychometric tests can be influenced by a number of variables, including but not limited to age, educational attainment, and the potential impact of learning effects. HE may go undiagnosed if these variables are disregarded. In order to overcome this issue, we investigated the potential of rest-fMRI with a BOLD echo-planar imaging (EPI) technique to assess brain functional connectivity in order to detect cognitive impairment related to the presence of HE in the analyzed pediatric patient. Moreover, the possibility of cognitive improvement following the surgical correction of the disease using Meso-Rex surgery, as described in [47], is also investigated.

##### Characteristics, Data Acquisition, and Preprocessing

The patient, an 8-year-old boy, was admitted with cognitive impairment characterized by psychomotor sluggishness, a decline in fine motor performance, attention deficits, and a profound reduction in the ability to perform daily tasks. Standard liver function tests and hematologic markers were determined by obtaining and analyzing blood samples from veins using conventional methods. A measurement of venous ammonia confirmed the presence of ammonia.

The patient underwent Doppler ultrasonography, magnetic resonance (MR) imaging, and MR angiography to diagnose and assess portal cavernoma, collaterals, and spontaneous shunts. Specifically, baseline MR imaging (MRI) examinations were performed on a 3T MRI scanner (Discovery 750w, General Electric Medical System, Milwaukee, WI, USA) utilizing a 32-channel head coil during PRE, i.e., before the surgical correction of the portal cavernoma by means of Meso-Rex surgery and during two follow-up phases, i.e., 1 month (POST1) and 12 months (POST12) after the surgical treatment. The subject was positioned in the scanner with his head comfortably restrained by foam padding to reduce head movement. Earplugs were used to reduce the noise of the scanner. During the resting-state scan, the subject was instructed to keep his eyes closed, remain as motionless as possible, and clear his head of any particular thoughts. A standard multi-parametric MRI protocol was carried out with fast spin-echo (FSE) T1-weighted and T2-weighted MR images, fluid-attenuated inversion recovery (FLAIR), T2*-weighted gradient-recalled-echo (GRE), susceptibility-weighted imaging (SWI), and standard three-direction diffusion-weighted imaging. Isotropic T1-weighted volumetric imaging (3D-SPGR or MPRAGE) was acquired as anatomical reference images for rest-fMRI using a BOLD EPI technique, which was then performed to assess spontaneous neuronal activity in the resting state networks and evaluate brain network connectivity [2,15,33,48].

The volume of T1-weighted morphological data and functional slices, obtained respectively through MR and BOLD imaging, was appropriately preprocessed following a series of steps. First, morphological scans were preprocessed by correcting motion artifacts. The original data volume was transformed and normalized to the standard EPI template in the Montreal Neurological Institute (MNI) atlas (https://brainmap.org/training/BrettTransform.html, accessed on 28 September 2020) and restored to 3×3×3 mm${}^{3}$. The resulting images were spatially smoothed with an 8-mm full-width at half-maximum Gaussian kernel. Nonbrain tissues were removed from the scans, and a segmentation of the brain tissues was performed. Atlas-based cortical parcellation was obtained, and seed selection was carried out using Brodmann areas (BAs) (https://www.brainm.com/software/pubs/dg/BA_10-20_ROI_Talairach/functions.htm, accessed on 6 October 2022) after transforming the co-ordinates from the MNI atlas into the Talairach atlas (https://brainmap.org/training/BrettTransform.html, accessed on 6 October 2022). Then, confounds, i.e., noise variables representing fluctuations of a non-neuronal origin, such as residual physiological effects derived from subject motion, were estimated. These confounding effects were minimized by performing the so-called denoising procedure. To this aim, the CONN toolbox (https://web.conn-toolbox.org/, accessed on 11 October 2022) was used, which is an open-source MATLAB/SPM-based cross-platform software (https://www.nitrc.org/projects/conn, accessed on 11 October 2022). The CompCor function in CONN was used for spatial and temporal preprocessing to minimize the impact of motion and physiological noise factors, as well as to define and remove confounds in the BOLD signals. A regression of first-order derivative terms for the whole brain, ventricular, and white matter signals was also included in the correlation preprocessing to reduce the influence of spurious variance on neuronal activity.

##### Resting State Networks Identification

The assessment of brain functional connectivity for this patient was obtained for a given number of RSNs selected through a seed-based correlation approach. A seed region of interest (ROI) was first identified, and then the linear correlation of the seed region with all the other voxels of the entire brain was computed, making use of statistical analysis [12,14,49]. Among the commonly known and analyzed 36 RSNs [50], this procedure, for which we refer the reader to [10,49] for the details, allowed for the identification of eight resting-state networks with the best signal-to-noise ratio, following the fMRI image denoising and realignment steps. The chosen networks were then used to evaluate subject-specific cognitive fingerprints at the baseline and after disease correction and to show any significant improvement in the individual functional connectivity after surgery. All ROIs encompassing the eight selected RSNs (Default Mode—DM, SensoriMotor—SM, Visual—VS, Salience—SAL, Dorsal Attention—DA, FrontoParietal—FP, Language—L, Cerebellar—CB) were imported into the CONN Toolbox and then used to perform the seed-based extraction of Q=32 BOLD fMRI time series as sequences of M=200 consecutive synchronous values, considered as a realization of the network $S=\{S_{1},\ldots ,S_{Q}\}$ describing the neural dynamics.

##### Data and Statistical Analysis

Linear models in the form of (Equation 5) were fitted on each pair of BOLD time series $x=s_{i}$ and $y=s_{j}$ (i,j=1,…,Q, i≠j), preprocessed by removing the mean value and scaled to have a unitary standard deviation, for which the time-domain MI was then obtained as a measure of pairwise functional connectivity. In each experimental condition, the existence of every pairwise link was evaluated applying the surrogate data analysis and assessing the significance of the estimated MI, as detailed in Section 2.1.3.

Furthermore, the OI measure was computed for a predefined number of multiplets from order N=3 to order N=8. Specifically, among all the possible combinations of order 3 derived from the Q=32 time series, 56 triplets were selected randomly from different RSNs. These triplets were then used as roots for building 40 multiplets of order 4, where the additional time series was chosen randomly within the remaining RSNs. The procedure was iterated for higher orders, eventually obtaining 30 multiplets of order 5, 20 multiplets of order 6, 5 multiplets of order 7, and 1 multiplet of order 8. For each order and multiplet, the significance of the estimated OI and OI increments was assessed by applying the bootstrap method, as detailed in Section 2.1.3. Specifically, we set L=50 for the generation of the bootstrap fMRI time series of length M=200. Moreover, the significance of the differences between the MI/OI values measured in three conditions (PRE vs. POST1, PRE vs. POST12, and POST1 vs. POST12) was assessed by comparing the MI/OI distributions obtained through the block bootstrap method, as detailed in Section 2.1.3.

## 3. Results and Discussion

This section displays and interprets the results obtained for the simulation example of Section 2.2.1 (Section 3.1) and the application to the clinical case presented in Section 2.2.2 (Section 3.2).

### 3.1. Simulation Example

The results of the analysis relevant to the theoretical simulation of Section 2.2.1 are shown in Figure 2. The MI values shown in Figure 2B reflect the strength of the relationships between pairs of variables; the values of MI >0.5 nats are found for the pairs $\left\{X_{2},X_{3}\right\}$, $\left\{X_{2},X_{6}\right\}$, $\left\{X_{3},X_{6}\right\}$, $\left\{X_{3},X_{7}\right\}$, and $\left\{X_{6},X_{7}\right\}$. However, not all of these connections are true links of interaction between the investigated variables, as happens, e.g., for the pairs $\left\{X_{2},X_{6}\right\}$ and $\left\{X_{6},X_{7}\right\}$. Indeed, the nodes $X_{2}$ and $X_{6}$, as well as $X_{6}$ and $X_{7}$, are not linked by direct interaction pathways but still show nonzero connectivity (Figure 2A). This finding is related to the existence of common driving and chain effects in these cases, respectively, which determine the appearance of indirect links of interaction between the two investigated variables [31]. This misinterpretation of the network structure does not occur in the case of the common child effect since truly nonsignificant MI is found for the pairs $\left\{X_{1},X_{4}\right\}$, $\left\{X_{1},X_{5}\right\}$, and $\left\{X_{4},X_{5}\right\}$, as shown by the absence of links between these variables in Figure 2B.

A high-order representation of the investigated interactions is provided in Figure 2C,D. The OI values in Figure 2C show an expected increase in redundancy as the network size increases (i.e., from order 3 to 7), even though some synergistic multiplets are still found at orders 3 and 4. Specifically, as shown by the values of the OI increment in Figure 2D, the synergistic triplets (first column, order 3) are those containing the variables $X_{1}$, $X_{4}$, and $X_{5}$, which, indeed, are involved in the common child structure (Figure 2A). In addition, the chain structure for which the node $X_{3}$ is a sink for $X_{2}$, and the same applying for $X_{6}$ and $X_{7}$ with respect to $X_{3}$ (Figure 2A), causes the synergy that also involves these variables when combined with $X_{1}$, $X_{4}$ or $X_{5}$. Interestingly, this pattern is maintained at higher orders, with most of the multiplets comprising the variables $X_{1}$, $X_{4}$, and $X_{5}$, for which significant synergistic OI increments are found. On the other hand, the triplets $\left\{X_{2},X_{3},X_{6}\right\}$, $\left\{X_{2},X_{3},X_{7}\right\}$, and $\left\{X_{3},X_{6},X_{7}\right\}$, along with others, such as $\left\{X_{1},X_{2},X_{3}\right\}$, $\left\{X_{1},X_{2},X_{6}\right\}$, and $\left\{X_{1},X_{2},X_{7}\right\}$, show positive values of the OI increment (Figure 2D, first column, order 3), confirming that the common driver and chain structures are dominated by redundancy (Figure 2A). As happens for the synergistic variables $X_{1}$, $X_{4}$, and $X_{5}$, the addition of the variables $X_{2}$, $X_{3}$, $X_{6}$, and $X_{7}$ to form groups of orders 4, 5, 6, and 7 is likely to significantly increase the redundancy of the interactions within the network, as shown by the red squares containing these variables in Figure 2D (the second, third, fourth, and fifth column).

The bootstrap data approach, applied to the simulated time series to retrieve confidence intervals for the proposed measures, allowed us to statistically validate the OI values and the OI increments (Figure 2C,D). Specifically, in Figure 2C, the nonsignificant OI values are depicted as grey circles around the zero threshold; at order 3, the number of nonsignificant OI values is the highest (4 over 35 multiplets). Conversely, in Figure 2D, the nonsignificant OI increments are shown as grey squares, where each square corresponds to the target-specific OI increment for that multiplet. The nonsignificant Δ(OI) values are found at orders N=4,5, especially when the targets $X_{4}$ and $X_{5}$, as well as $X_{6}$, are added to form multiplets containing the variables $X_{2}$, $X_{3}$, and $X_{7}$ or $X_{6}$.

In conclusion, this simulation example shows that the connectivity maps traced by the MI do not provide a fully explanatory description of the complex and multiple interactions taking place in the analyzed network. Indeed, different network structures, such as common driver, chain, and child ensembles, are not always truly reproduced by these pairwise maps, and the resulting MI values between the pairs of observed variables may be biased. The utilization of high-order measures investigating the relationships between more than two variables is fundamental to provide a more complete description of the connectivity patterns emerging from the network. It is noteworthy that the possibility to specify the redundant and/or synergistic character of groups of variables would allow for a more faithful representation of the network ground structure. Moreover, the use of surrogate and bootstrap methods is essential for disregarding nonsignificant pairwise and high-order connectivity links between the observed variables.

### 3.2. Application to Brain Networks

The results of the analysis relevant to the application of the proposed methodology to the clinical case discussed in Section 2.2.2 are reported in Figure 3, showing the subject-specific maps of brain functional connectivity in the three conditions (Figure 3A), the distributions of the OI values for all the orders (N=3,…,8), where each order comprises a given number of analyzed multiplets (Figure 3B), and the values of the OI increment computed, as in (Equation 3), for each target $s_{j}$ within the multiplet $s^{N}$ at order *N* (Figure 3C).

The presurgery phase is characterized by a relatively sparse functional connectivity network (Figure 3A), with 41% of the MI values detected as statistically significant by surrogate data analysis. When compared to this phase, the immediate postsurgery period shows a weakening of functional brain connectivity, as evidenced by the lower number of statistically significant MI values between the pairs of BOLD series determined by the surrogate data approach (Figure 3A, PRE vs. POST1). Indeed, the global density of the network, i.e., the number of significant connections, decreases from 41% in PRE to 25% in POST1. The number of significant connections increased markedly 12 months after surgery (Figure 3A, POST12, 52% of significant connections), suggesting that the proposed surgery correction of the portosystemic shunt worked in recovering brain functional connectivity in this patient. Interestingly, the local densities, i.e., the number of significant functional connections within (and between) RSNs, are characterized by a drop in the immediate postsurgery period (Table 1, POST1) followed by an increase 12 months after the treatment (Table 1, POST12), for almost all the (pairs of) RSNs. This suggests that the weakening and reduction in the number of links within the network is not localized to a specific brain area but spread over the whole cortex.

The decrease in functional connectivity 1 month after the treatment and its increase 12 months after, observed with the pairwise estimator of MI, are translated into consequent decreases and increases in the OI values, respectively. Specifically, while all three phases are characterized by the presence of a great number of nonsignificant connections, the strength and number of these high-order links decreased 1 month after and increased again 12 months after the treatment (Figure 3B; e.g., in the case of N=3, the significance rate goes from 16% in PRE to 3.6% and 16% in POST1 and POST12, respectively). This finding confirms the main result coming from MI analysis, i.e., that the surgery correction of the portosystemic shunt worked in recovering brain connectivity in this patient. Moreover, in the last experimental phase, the number of synergistic interactions predominates over redundancy, suggesting that the recovered brain can display synergy as an emergent behavior, as well as that synergistic interactions may serve to integrate and complement redundant subnetworks in recovered physiological conditions.

These findings are confirmed by the decrease in Δ(OI) values in POST1, and their subsequent increase in POST12 (Figure 3C, POST1 vs. POST12), characterized by a tendency towards synergy (dark-blue rectangles).

Figure 4 shows the maps of the pairwise differences between the MI (Figure 4A) and OI (Figure 4B) values computed in two different experimental conditions. The impairment of brain connectivity in POST1 and its recovery in POST12 is confirmed for this patient, looking at the variations in the MI across conditions established by the bootstrap technique (Figure 4A). This analysis documents, indeed, that the functional connectivity decreases 1 month after surgery (red squares in POST1-PRE) but markedly increases 1 year after (green squares in POST12-PRE and POST12-POST1), suggesting an improvement in cognitive functions for this patient. The utilization of the bootstrap technique for the detection of OI variations across conditions are confirmed in the results shown in Figure 3B,C. In detail, the high number of nonsignificant differences, indicated by grey rectangles in Figure 4B, reflect the presence of nonsignificant OI values in the three conditions (Figure 3B, grey circles) with only a few significant connections left. Looking at the differences between the experimental conditions, the decrease in OI values toward synergy 12 months after the treatment is documented by the predominance of red rectangles in Figure 4B, POST12-PRE and POST12-POST1. This reduction is localized to specific multiplets, suggesting that the recovery of high-order interactions is specific to certain areas of the pediatric brain.

Our preliminary results agree with previous findings [38], obtained by applying multivariate information metrics to fMRI data and documenting the presence of copious and widely distributed synergistic subsystems across the entire cerebral cortex. In our application to fMRI data, we randomly selected nodes from different RSNs to build high-order structures comprising between 3 and 8 regions and showed that synergistic subsets are ubiquitous, arising at higher orders systematically across the cortex. Specifically, while redundant interactions dominate at larger subset sizes, especially during the pre-operative and the immediate post-operative phases, the late post-operative phase is characterized by the appearance of a previously hidden repertoire of synergistic ensembles, as also demonstrated by the bootstrap data analysis applied to detect subject-specific differences between conditions. In detail, these randomly sampled assemblies expressing synergy were found to involve nodes from the DM, SAL, and FP networks in the pre-operative phase, together with the VS, L, and CB networks twelve months after the surgical correction, when the number of synergistic pathways of interaction was definitely increased.

The application of multivariate information measures demonstrates that high-order synergies represent a kind of “shadow structure” emerging from resting-state brain activity and missed by bivariate functional connectivity approaches, which indeed reveal redundancy-dominated correlations and do not provide an overall map of the statistical structure of the network [36,37,38]. Given the novelty of our findings, the significance of these synergistic dependencies remains almost entirely unknown, although the clinical importance of studying and comprehending these intriguing patterns persists unaltered.

In conclusion, we propose here a subject-specific statistical evaluation of functional network connectivity analysis in the peculiar case of pediatric portal cavernoma. Our mathematical framework, based on the combined use of pairwise and high-order functional connectivity measures, allowed us to display subject-specific features of brain connectivity in a patient before and after the surgical correction of the portosystemic shunt. Moreover, the utilization of surrogate and bootstrap data analyses was essential to statistically validate the functional connectivity maps obtained before surgery and during the follow-up phases (1 month and 12 months after the surgical treatment), as well as the differences between pairs of these. This has great clinical relevance for single-subject investigations and treatment planning, particularly when it is necessary to study the effects of clinical diseases on single individuals and the subject-specific responses to personalized diagnosis and care. The statistical assessment of intra-subject connectivity network changes over time could be interpreted as evidence of statistically significant increases/decreases in functional connectivity related to an event, i.e., the surgical procedure to remove the shunt in our clinical case. Specifically, the overall increase in the number and strength of functional pairwise and synergistic connections after the surgical treatment, resulting from our analyses, was confirmed by clinical findings: during the follow-up phases, the patient recovered well from HE, as evidenced by the improvement in his cognitive functions, the recovery from psychomotor sluggishness and attention deficits, and the subsequent return to school, which he had dropped out of before the treatment. Therefore, the proposed statistical approaches can successfully help scientists and clinicians to identify significant pairwise but especially high-order brain functional connectivity signatures on a single-subject basis in different physiological and diseased conditions.

## 4. Conclusions

In this study, we propose a subject-specific statistical assessment of pairwise and high-order functional connectivity in brain networks, thus relying on a dual aim. First, our work supports the use of surrogate and bootstrap data analyses for the single-subject investigation of brain connectivity in fMRI studies. Moreover, it expands the well-established framework of pairwise functional connectivity analysis to the less explored concept of high-order interactions in brain networks, which allows for uncovering effects and connection modalities that, otherwise, using the current pairwise approaches, would remain hidden. In perspective, the proposed single-subject analysis may have clinical relevance for subject-specific investigations and treatment planning. Indeed, the method based on surrogate and bootstrap data generation is able to reproduce the peculiar features of brain network connectivity on a single-subject basis. While this approach should be tested on a larger number of individuals to validate the clinical findings, it still revealed clinically and physiologically plausible patterns of brain pairwise connectivity in the reported application. On the other hand, the possibility to investigate brain connectivity and its post-treatment functional developments at a high-order level was essential to fully capture the complexity and modalities of the recovery. The results obtained here, albeit in a preliminary fashion, support the need to investigate the complex behavior of brain structures and their emergent synergistic patterns. We assert that high-order interactions in the brain represent a vast and under-explored space that is now accessible using the tools of multivariate information theory, and this may offer novel scientific insights, even in today’s clinical practice. In conclusion, this study paves the path for extensive examinations on larger datasets to assess the coherence between subject-specific observations and their clinical significance in a broader patient population. Future developments should further address the importance of combining multiple functional connectivity methods to achieve a thorough description of brain networks. Additionally, these advancements should also consider transient behaviors [51], enhance the identification of topological and causal structures [31,52], and go beyond the use of the first-order gradient (i.e., the OI increment defined in (Equation 3)). Expanding the study in terms of increments in information across orders, as well as of their assessment through surrogate and bootstrap approaches, would allow researchers to further unveil synergistic structures and investigate their role in complex networks of multiple interacting nodes in the brain [53]. Furthermore, the exploration of dynamic forms of pairwise and high-order connectivity [18,54], which account for temporal correlations in the detection of brain functional couplings, may spark great interest among neuro-scientists and assume high relevance in the field of fMRI data analysis.

## Figures and Tables

**Figure 1 life-13-02075-f001:**
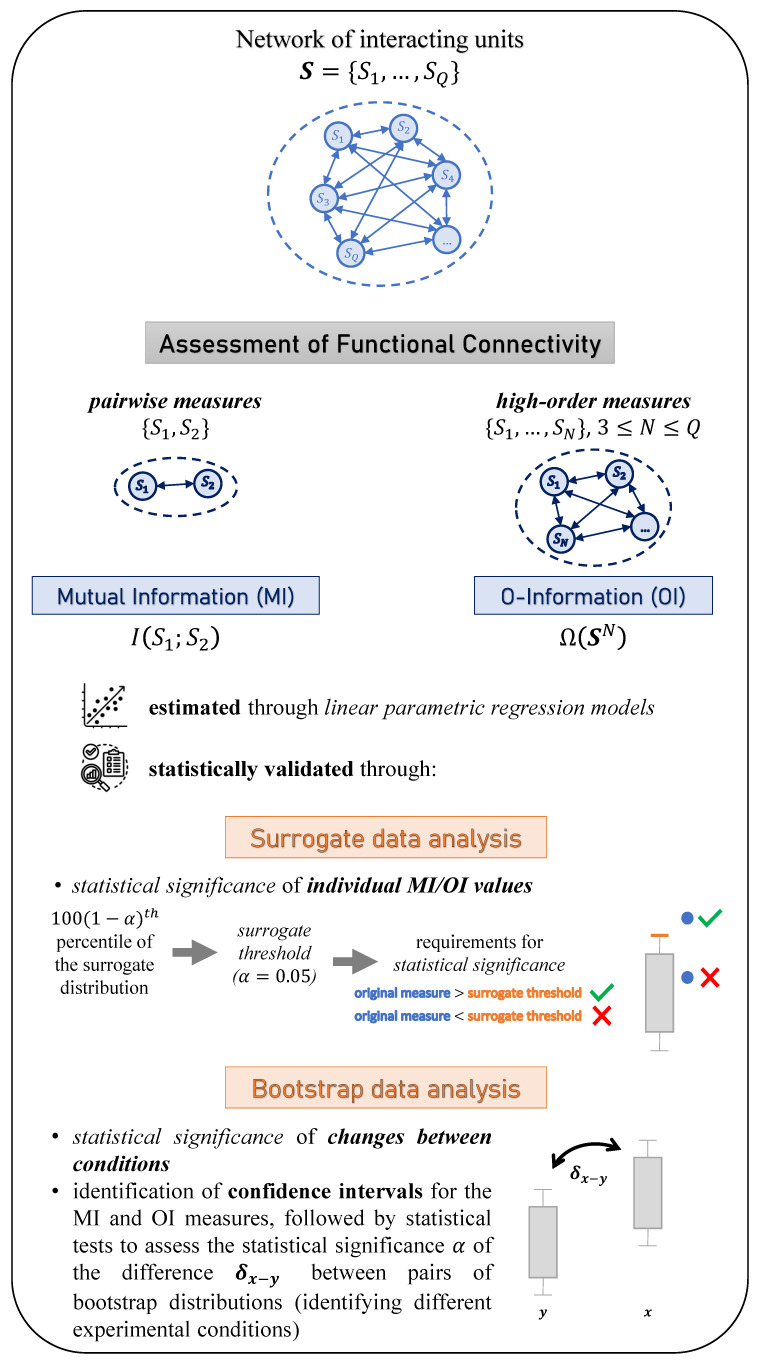
Schematic workflow of the proposed methodology.

**Figure 2 life-13-02075-f002:**
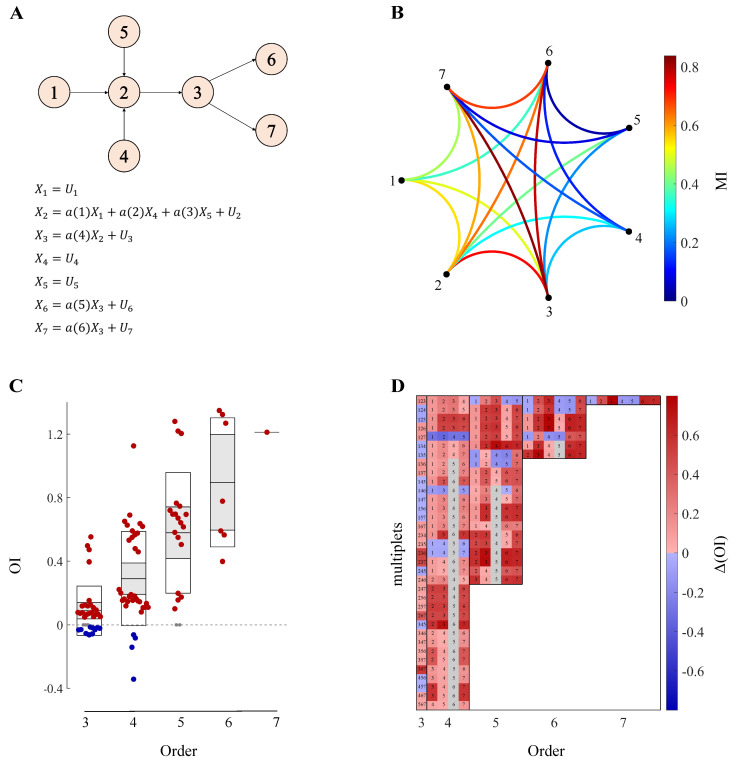
Theoretical simulation shows that high-order measures capture the synergistic and redundant characters of the interaction pathways involving multiple variables within complex networks, as well as the fact that the surrogate and bootstrap methods are helpful in the detection of significant interaction pathways among those variables. (**A**) Simulation design (**top**), where the numbers identify the variables, and model equations (**bottom**), where $U_{i}$, i=1,…,6 is the zero-mean random noise with unit variance. (**B**) Circular graph representing the MI-weighed significant connections among pairs of simulated variables. Non-significant links detected through surrogate data analysis are not drawn. (**C**) Boxplots representing the distributions of the OI values for all the multiplets from order 3 to 7. In each box, the central black mark indicates the mean, and the bottom and top edges of the box indicate the 25th and 75th percentiles, respectively; the red, blue, and grey circles indicate positive (redundant), negative (synergistic), and nonsignificant OI values, respectively. (**D**) Δ(OI) values computed for each target (numbers in the squares) inside the multiplets (sequences of numbers along each row) at all orders (separated by black vertical lines). The red, blue, and grey squares indicate positive (redundant), negative (synergistic), and nonsignificant Δ(OI), respectively, brought by that target to the multiplet for a given order. The values of OI and Δ(OI) are expressed in nats, i.e., natural units.

**Figure 3 life-13-02075-f003:**
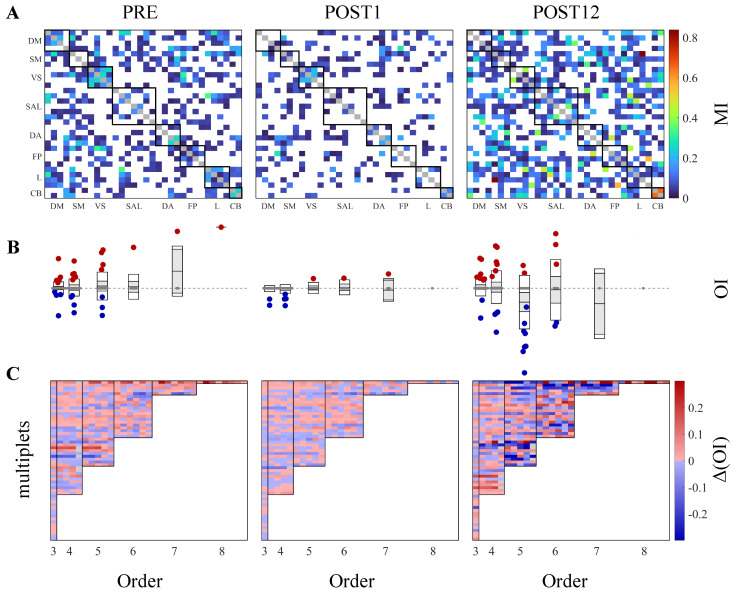
In the application to resting-state fMRI data, the MI detects an increase in the number and strength of connections, while the OI reveals high-order synergistic interactions 12 months after the surgical treatment. (**A**) Symmetric matrices representing the MI-weighed significant connections among pairs of variables in PRE (**left**), POST1 (**middle**), and POST12 (**right**) conditions. White spaces indicate nonsignificant connections. Black squares along the main diagonal group show all the time series belonging to the same RSN. (**B**) Boxplots representing the distributions of the OI values for all the multiplets from order 3 to 8, in PRE (**left**), POST1 (**middle**), and POST12 (**right**) conditions. In each box, the central black mark indicates the mean, and the bottom and top edges of the box indicate the 25th and 75th percentiles. The red, blue, and grey circles indicate positive (redundant), negative (synergistic), and nonsignificant OI values, respectively. The dashed grey horizontal line corresponds to the zero level. (**C**) Δ(OI) values computed for each target inside the multiplets at all orders (separated by black vertical lines). Red, blue, and grey squares indicate positive (redundant), negative (synergistic), and nonsignificant Δ(OI), respectively, brought by that target to the whole multiplet for a given order. Values of MI, OI, and Δ(OI) are expressed in nats, i.e., natural units.

**Figure 4 life-13-02075-f004:**
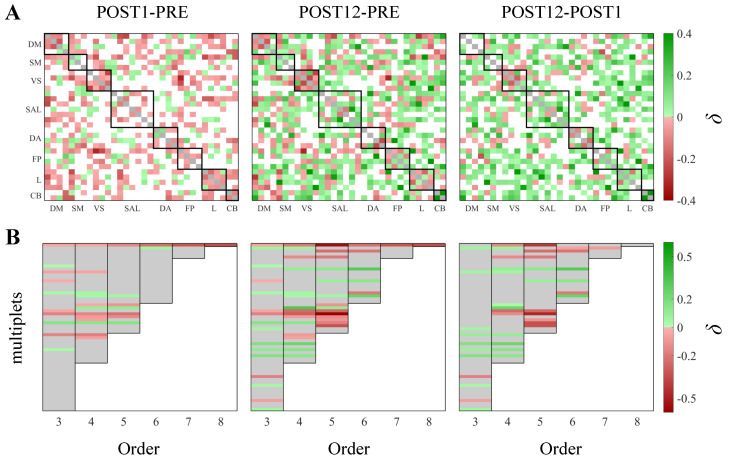
The difference in the MI and OI values between pairs of experimental conditions was assessed through the bootstrap data analysis. The figure shows networks of the differences (δ) between the MI (**A**) and OI (**B**) values estimated in two conditions; the green and red squares indicate positive and negative differences, respectively, while the white squares (**A**) or grey rectangles (**B**) indicate nonstatistically significant differences. In (**A**), the black squares along the main diagonal group represent all the time series belonging to the same RSN. In (**B**), the orders are separated by black vertical lines; for each order, the rows correspond to the multiplets selected for that order.

**Table 1 life-13-02075-t001:** Local density within and between resting state networks, expressed in % of significant connections, before the treatment (PRE, left) and 1 month (POST1, middle) and 12 months (POST12, right) after the treatment.

PRE									POST1									POST12								
	**DM**	**SM**	**VS**	**SAL**	**DA**	**FP**	**L**	**CB**		**DM**	**SM**	**VS**	**SAL**	**DA**	**FP**	**L**	**CB**		**DM**	**SM**	**VS**	**SAL**	**DA**	**FP**	**L**	**CB**
**DM**	50	67	25	54	31	38	69	38		16	25	25	18	31	38	31	13		50	58	50	46	38	44	44	63
**SM**		0	58	52	67	33	58	33			33	33	33	58	33	25	17			67	42	71	67	58	67	33
**VS**			100	14	25	69	31	25				67	25	25	6	19	25				50	43	31	63	63	75
**SAL**				33	29	36	39	21					14	29	18	25	7					62	32	50	57	64
**DA**					33	25	31	38						50	13	0	0						33	63	50	38
**FP**						50	13	25							17	19	25							50	50	38
**L**							83	63								50	13								67	50
**CB**								100									100									100

## Data Availability

Data available on request from the authors.

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
