# Peer review of "Statistical Approaches to Identify Pairwise and High-Order Brain Functional Connectivity Signatures on a Single-Subject Basis"

_life, 2023, doi:10.3390/life13102075_

Round 1

Reviewer 1 Report

In the article, the authors discussed the necessity to implement new methodologies for assessing the value of single-subject fingerprints of brain functional connectivity using standard pairwise and also novel high-order measures. The topic is interesting since new neuroscience methods need to be designed and studied.

However, there are a few points to be addressed before the publication:

2.     I would suggest inserting a workflow for better understanding the dynamics of the work. The only use of the equations seems to be hard to follow and somehow confused.

3.     I believe that the introduction part may be improved by adding more details about the rationale of the work and the authors’ hypothesis.

4.     Page 6: “The parameters a(i), i = 1, . . . , 6, 259 quantifying the pairwise relationships between the observed variables, are chosen in the 260 range [0.95 − 1], setting a(1) = a(2) = 0.99, a(3) = a(6) = 1 and a(4) = a(5) = 0.95”. Why did the authors choose those parameters? Please consider including more technical details in the manuscript.

5.     Another important point is the choice of the brain networks to study. Why did the authors choose these 8 networks? Did they use the standard ones or there is a rationale behind it? This part may be improved and included in the introduction or better discussed in section 5.

6.     What implications or improvement this method may have in the neuroscience and rs-fMRI analysis fields? And how it can help to overcome the current methodologies? These points may be improved in the conclusion.

Minor point:

I am not sure if these points are included in the “communications” type, but some parts as “Author Contributions” and “Acknowledgments” are missing.

 In some parts, the manuscript is difficult to follow. The overall grammar and the style of writing should be improved.

Author Response

We thank the reviewer for the overall positive evaluation of our work and for the useful comments. A point-to-point reply to the reviewer's answer is provided in the attached PDF document.

Reviewer 2 Report

The study proposes a subject-specific statistical assessment of pairwise and high-order functional connectivity in brain networks using fMRI data analysis. It aims to analyze individual brain signal recordings to extract meaningful insights about physiological and pathological states using statistical methods. It proposes a methodology to assess brain functional connectivity at both pairwise and high-order levels. The approach is validated in a pediatric patient with hepatic encephalopathy, showing potential clinical relevance. Understanding high-order brain connectivity is crucial for comprehending post-treatment functional developments accurately.

Major comments:

1. The scope of Life does not seem to match these statistical, simulation, and network neuroscience topics in other journals like Brain Sci or Clinical Investigation. Entropy or other statistical-based journals are far more appropriate.

2. The authors must specify their novelty. For example, it is not clear if the use of a single patient is the highlight of the study or a limitation. Does the novelty in the the theoretical basis for the statistical aspect or in the ‘new unexpected’ connectivity?

3. In the last decade, there have been 100 publications and 10 reviews articles published on brain functional connectivity networks. The authors mentioned some of them, it is important to add the highly cited review that states the challenges. Examples are Anderson et al. (2011) on functional connectivity "fingerprints" for individuals and populations by imaging. Rossini et al. (2019) in Clinical Neurophysiology; Cole et al. in Frontiers in Systems Neuroscience (2010), Wang et al. (2015) on the statistical methods of healthy and disease statistics, Wendling et al. (2009) on regression methods that showed sensitivity to the coupling parameter in all tested models combining measurements including EEG and magnetic resonance imaging (fMRI).

4. it is not clear what was the rational to use a patient with hepatic shunts that suffered from encephalopathy. The relevance of the connectome or other natural physiopathology in CNS diseases is not evident or expected in such case (the procedure it expected to increase/ decrease brain toxicity).

5. Some of the claims are difficult to accept without any further support and better not to over interpretate. or example, the physiological support for underlying lower or fewer connectivity 1-month after surgery and the statement that "markedly increased one year later ….suggesting an improvement of cognitive functions for this patient" are not supported by the relevant data. If data on this is available, please provide it.

6. The focus on eight brain regions is not discussed. It thus remains a bit mysterious to readers that are not familiar with brain anatomy and default connectivity.Maybe add in supplementary a schematic atlas with their regions. 

Minor comments

  1. It is better to structure the manuscript in a more routine format of Materials and Methods, Results Discussion, etc.
  1. Please explain what it means by significance rate.
  1. Please be more specific or remove fuzzy writing. For example, … the communication of sub-groups of variables, thus pertaining to information that is replicated across numerous elements of the complex system, can resolve uncertainty across all the other elements of that system". Is it statement apply to all complex systems (e.g. weather change) or it is for human brain?
  1. It makes more sense to define the "non-significant OI" (gray color) with a range of error interval.

The writing is clear and only minor editing is needed

Author Response

(The authors gave the same response as above.)

Round 2

Reviewer 1 Report

Thank you for answering and addressing all my comments.